# Deletion of LBR N-terminal domains recapitulates Pelger-Huet anomaly phenotypes in mouse without disrupting X chromosome inactivation

Alexander Neil Young[1,2], Emerald Perlas[1,6], Nerea Ruiz-Blanes[2,6], Andreas Hierholzer[1,5], Nicola Pomella[2], Belen Martin-Martin[2], Alessandra Liverziani[1], Joanna W. Jachowicz [4], Thomas Giannakouros[3] & Andrea Cerase [1,2✉]

Mutations in the gene encoding Lamin B receptor (LBR), a nuclear-membrane protein with sterol reductase activity, have been linked to rare human disorders. Phenotypes range from a benign blood disorder, such as Pelger-Huet anomaly (PHA), affecting the morphology and chromatin organization of white blood cells, to embryonic lethality as for Greenberg dysplasia (GRBGD). Existing PHA mouse models do not fully recapitulate the human phenotypes, hindering efforts to understand the molecular etiology of this disorder. Here we show, using CRISPR/Cas-9 gene editing technology, that a 236bp N-terminal deletion in the mouse *Lbr* gene, generating a protein missing the N-terminal domains of LBR, presents a superior model of human PHA. Further, we address recent reports of a link between *Lbr* and defects in X chromosome inactivation (XCI) and show that our mouse mutant displays minor X chromosome inactivation defects that do not lead to any overt phenotypes in vivo. We suggest that our N-terminal deletion model provides a valuable pre-clinical tool to the research community and will aid in further understanding the etiology of PHA and the diverse functions of LBR.

[1] EMBL-Rome, Epigenetics and Neurobiology Unit, Monterotondo (RM), Italy. [2] Blizard Institute, Centre for Genomics and Child Health, Barts and The London School of Medicine and Dentistry, Queen Mary University of London, London, UK. [3] Laboratory of Biochemistry, Department of Chemistry, Aristotelian University, Thessaloniki, Greece. [4] Division of Biology and Biological Engineering, California Institute of Technology, Pasadena, CA, USA. [5] Present address: Department of Biosystems Science and Engineering, ETH Zürich, Basel, Switzerland. [6] These authors contributed equally: Emerald Perlas, Nerea Ruiz-Blanes. ✉email: a.cerase@qmul.ac.uk

LBR is an inner nuclear-membrane protein, having a dual function in the cells in which it is expressed. It works as a sterol reductase in the cholesterol biosynthesis pathway and as nuclear tether for membrane proteins and heterochromatin organization[1]. Mutations in *Lbr* have been associated with two genetic disorders, Pelger–Huet (PHA, OMIM #169400) and Greenberg Dysplasia (GRBGD, #215140)[2–4]. Pelger–Huet Anomaly is a rare dominant, benign disorder, involving hemizygous *Lbr* mutations[5]. Its frequency is estimated to be between 0.01 and 0.1% of the population, although in some parts of the world this frequency can be considerably higher[2]. While this disorder is considered a benign blood anomaly, its association with severe blood disorders, such as acute lymphoblastic leukemia, has been reported[6]. PHA patients show the following phenotypes: hypolobulated neutrophil nuclei with clamped chromatin, hyposegmented granulocytes, with a sub-cohort of patients also showing skeletal defects such as syndactyly or polydactyly[7]. Homozygous mutations in *Lbr* are causative of Greenberg Dysplasia. GRBGD is embryonic lethal in human and the aborted fetuses present the following phenotypes: fetal hydrops, severe shortening of long bones, "moth-eaten" radiographic appearance, platyspondyly, disorganization of chondroosseous calcification, and ectopic ossification centers[8]. Interestingly, the complete loss of LBR in mouse gives rise to different phenotypes. Homozygous *Lbr* KO mice, show incomplete embryonic lethality[5], display some of the PHA phenotypes (such as blood defects and skeletal malformations), but also show severe fur/skin defects, such as alopecia, ichthyosis, and hydrocephalus phenotypes[9]. Considering the very high level of identity between the human and mouse LBR protein (~80%) and the conservation of the cholesterol biosynthesis pathways between these species[10], these differences cannot be fully explained by current literature data.

Recently *Lbr* has also been reported to have a role in X chromosome inactivation (XCI)[11]. Biochemical purifications of the proteins interacting with Xist RNA, the master regulator of XCI, reported LBR to be a specific and abundant Xist interacting protein[12]. Subsequently, Chen et al. showed that this interaction is needed for XCI, both in inducible Xist systems and differentiating female ESCs[11]. In particular, they showed that this interaction allows the efficient spreading of Xist RNA onto active genes and the proper localization of the future inactive X chromosome (Xi) to the nuclear periphery[11]. Considering the discrepancies between the human diseases and mouse model phenotypes, LBR's diverse and emerging cellular functions, novel mouse models are needed for a better understanding of its roles in development and XCI. For these reasons, we have generated and characterized a novel mouse model, allowing us to separate the activities of LBR N-terminal domains from the sterol reductase domain. We report that this model closely recapitulates the phenotypes of the human disease (PHA) and shows minor XCI defects.

## Results

### Generation of *Lbr* mutants

In order to generate a mouse model more closely recapitulating the human phenotype, and to study *Lbr* functions in XCI in vivo, we have decided to generate several mutations of this gene, by means of CRISPR/Cas9 technology, targeting LBR N-terminus (see "Methods"). We selected for mutations generating N-terminal truncations (i.e., of the domains implicated in the interaction with Xist RNA[11]), leaving intact the transmembrane sterol reductase domains. In line with these criteria, we decided to focus on a 236 bp deletion (*Lbr*[236] hereafter) spanning the canonical translation initiation site (ATG, exon2) and part of the *Lbr* first intron (Supplementary Fig. 1). This mutation disrupts the 3′ splicing signals in intron 1, resulting in aberrant intron 1 retention (Supplementary Fig. 1b). As the canonical, first translational initiation signals have been deleted in our mouse model, the translational machinery can only use downstream alternative translation start sites in the *Lbr* gene. An in silico analysis of our mutant Lbr mRNA suggests, with high confidence, Methionine 118 (M118) of the WT protein as a potential alternative translation start site. However, we cannot formally exclude the usage of other initiation start sites (see "Methods"). Western blot (WB) and immunoprecipitation–mass spectrometry (IP-MS) experiments support the bioinformatic prediction, suggesting that M118 is the most likely new translation start site in our mutant (Supplementary Fig. 1c). This deletion is expected to produce a truncated protein, lacking part of its globular domain and the entire LBR N-terminal RS and Tudor domains (N-terminal deletion, NT hereafter) (Supplementary Fig. 1c). These domains have been reported to interact with RNA (including Xist RNA) and HP1/peripheral heterochromatin, respectively[11,13]. Because this mutation generates a truncated but otherwise WT protein, our mutant allows us to separate LBR sterol reductase (transmembrane-domain) functions from XCI/nuclear periphery localization activities (N-terminal domains)[8] (Supplementary Fig. 1c, d).

### Characterization and analysis of *Lbr* mutants

Phenotypical analysis of our mouse model shows blood defects (laminopathies) (Fig. 1a) that are characteristic of the human PHA but not skin defects other than strain-associated sporadic dermatitis[14] (Jackson Lab, https://www.jax.org/, we do not observe ichthyosis) (Fig. 1b). We detected defects in the bone marrow blood precursors, suggesting maturation defects in neutrophil cells, as previously reported[15]. Homozygous NT mutant (NT-KO, Lbr[236]/Lbr[236]) mice do not show known skeletal malformations (such as syndactyly or polydactyly)[8] (Fig. 1c). We do not observe any significant weight difference between the wild-type (WT), heterozygous (HET, Lbr[+]/Lbr[236]), and *Lbr*[236] homozygous (Lbr[236]/Lbr[236]) adult animals (30.089 g ± 4.41; 32.05 g ± 2.58; 30.775 g ± 6.06, respectively). Mutations in the LBR sterol reductase domain, in homozygosis, are causative of GRBGD phenotypes[8]. As expected, we do not find Greenberg dysplasia phenotypes in our mouse model. However, we do find ~35% perinatal death in *Lbr*[236] homozygous crosses (adjusted $p = 0.0138$) (Fig. 1d and "Methods"), in agreement with Hoffman and colleagues[2]. We also report a slightly-increased embryonic lethality from heterozygous crosses, when the mutation is inherited from the father in heterozygosis (~26%) (Fig. 1d)[16,17]. We observed a notable difference in the average litter size between WT and NT-KO animals (3.2 animals/litter in *Lbr*[236] homozygous crosses vs 5.5 animals/litter in WT/WT crosses). Analysis of males and female gonads shows no evident defects, further suggesting elevated embryonic lethality in our mouse model, over fertility defects (Supplementary Fig. 2), as previously reported[5,9].

### Role of *Lbr* in development and XCI in vivo

To test the role of *Lbr* in XCI in vivo, considering the benign autosomal-dominant inheritance pattern of the disease, we set up several crosses with a particular focus on heterozygous crosses (HET, Lbr[+]/Lbr[236]) (Fig. 2a). From these crosses, the expected outcome is ~25% of LBR NT-KO (*Lbr*[236]/*Lbr*[236]) animals, equally divided between the sexes. We anticipated seeing a significant skewing against females, if *Lbr* was to play a major role in random XCI[11]. We also hypothesized a complete loss of female NT-KO animals from these crosses, in the case that the LBR N-terminal domains are critical factors for XCI in vivo, as shown for *Xist*[18] in random and

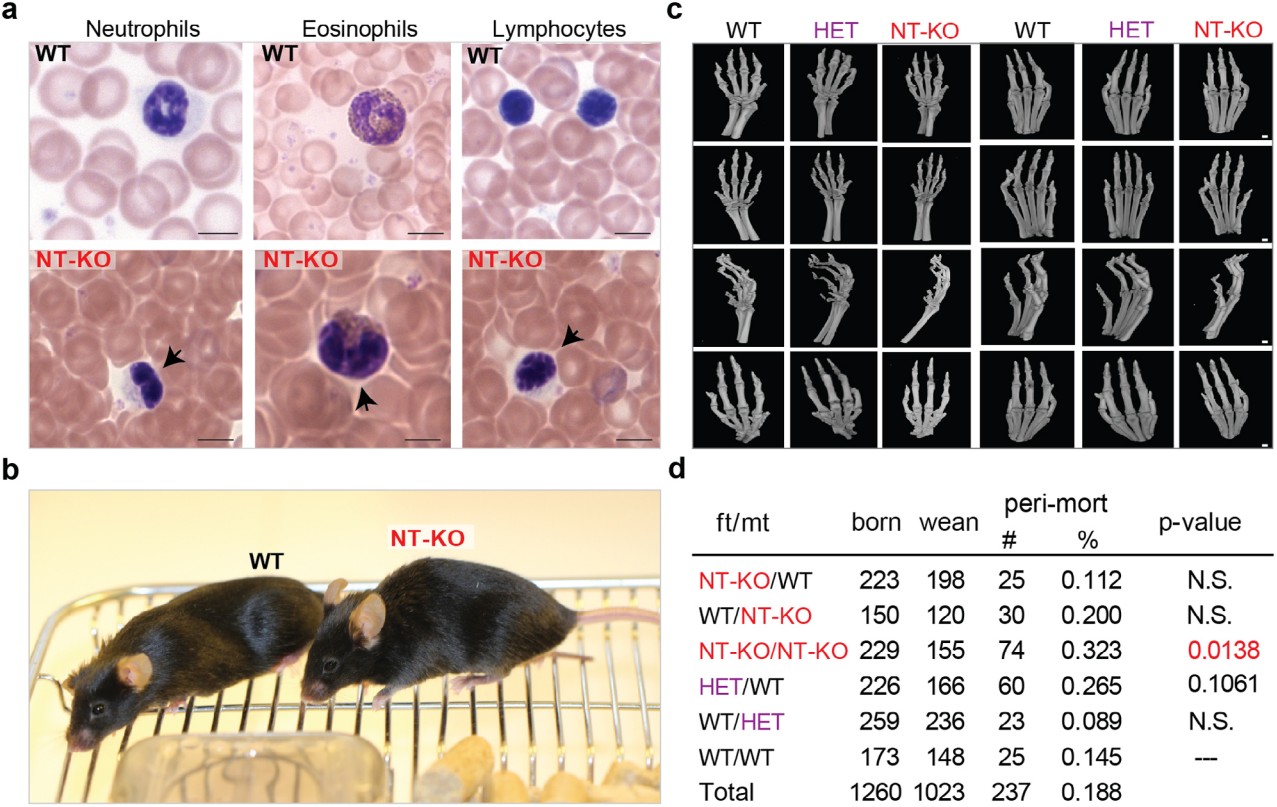

**Fig. 1 LBR NT-KO mice show laminopathy defects and increased perinatal mortality but not skin or skeletal defects. a** Top, Pelguer–Huet characteristic neutrophil and eosinophil and lymphocyte chromatin defects are shown (blood from female animals are presented, but no sex-specific effect has been observed). Black arrows in LBR NT-KO animals indicate defects in chromatin organization in the shown cell types. Scale bar indicates 5 μm. **b** No skin/fur defects were observed in LBR NT-KO animals. A female adult WT and a LBR NT-KO mouse are shown next to each other are shown. **c** No skeletal defects in front (left) and rear (right) paws have been observed in any analyzed category (female animals are shown in this figure, but no sex-specific effect has been observed). Scale bar indicates 1 mm. **d** The table shows perinatal mortality from different classes of crosses indicating the parental origin of the mutation (father: ft; mother: mt). born: born; wean: weaned; peri-mort: perinatal mortality; mutation classes are shown. A one-way ANOVA (Kruskal–Wallis test), followed by a post hoc multiple comparison tests (Dunn's test) have been used to compare the number of weaned mice in each group to the WT/WT condition (the analysis was done per litter).

| ft/mt | born | wean | peri-mort # | peri-mort % | p-value |
|---|---|---|---|---|---|
| NT-KO/WT | 223 | 198 | 25 | 0.112 | N.S. |
| WT/NT-KO | 150 | 120 | 30 | 0.200 | N.S. |
| NT-KO/NT-KO | 229 | 155 | 74 | 0.323 | 0.0138 |
| HET/WT | 226 | 166 | 60 | 0.265 | 0.1061 |
| WT/HET | 259 | 236 | 23 | 0.089 | N.S. |
| WT/WT | 173 | 148 | 25 | 0.145 | --- |
| Total | 1260 | 1023 | 237 | 0.188 | |

imprinted XCI, or for *Rnf12* in imprinted XCI[19]. From these crosses, on the contrary to our predictions[11], we observed no statistically-significant bias against female LBR NT-KO offspring (Fig. 2a, b). A detailed analysis of the proportion of male and female (m/f) offspring from different types of crosses (i.e. when the mutation is maternally or paternally inherited in heterozygosity), also revealed no significant differences[19] (Fig. 2b).

In order to understand the lack of an XCI phenotype, we proceeded to test whether the inactive X-lamina co-localization was affected in our mutants, as previously reported[11] (Fig. 2c). We used H3K27me3 as a surrogate marker of the inactive X chromosome (Xi)[20]. By immunofluorescence analysis, we could find no statistically significant changes in the Xi localization at the nuclear lamina in mouse tissues highly expressing *Lbr*, such as the thymus in post-weaning animals (WT: 1.649 ± 0.69 μm; NT-KO: 1.703 ± 0.82 μm, Fig. 2c). Similarly, RNA-seq analysis on tissues expressing *Lbr* such as liver, collected from post-weaning mice (WT, NT-KO), revealed no XCI phenotypes. Surprisingly, gene deregulation defects were more prominent in male (NT-KO vs WT; 209 differentially expressed genes (DEGs), FDR ≤ 0.05, absolute fold change log2 >1; 98 downregulated and 111 upregulated), than female (NT-KO vs WT; 12 DEGs, FDR ≤ 0.05, absolute fold change log2 >1; 6 downregulated and 6 upregulated (Fig. 2d and Supplementary Data 1). GO analysis revealed common defects in the membrane/endomembrane

organization, immune system, and protease inhibition as previously reported[21]. In particular, significantly deregulated genes further supports immune system and endomembrane defects, both in the nuclear envelope and the endoplasmic reticulum (Supplementary Fig. 3). Our observations are consistent with previous data since LBR was shown to diffuse between the nuclear envelope and the endoplasmic reticulum[22].

**Role of *Lbr* in random XCI**. In differentiating mouse embryonic stem cells (ESCs), *Lbr* plays a critical role during the initial spreading of Xist RNA onto active genes and subsequent stabilization of silencing through localization at the nuclear periphery[11,23], though its role in gene silencing has been debated[24]. Xist RNA was shown to interact with LBR via its N-terminal RS domain[11,25], which is expected to be missing in our animal model. In order to test LBR NT-KO role in early phases of random X chromosome inactivation, we derived female ES cells from Lbr homozygous crosses (*Lbr*[236]/*Lbr*[236]) crosses and matched WT animals. To determine the best timepoint for analysis, we differentiated WT cells for 0, 3, and 5 days (see "Methods"). As at day 3 of differentiation only a minority of cells have initiated XCI (~40%), we decided to do the analysis at day 5 of differentiation, where 70–80% of the cells have initiated random XCI. In brief, we differentiated matched WT and LBR NT-KO females ESCs and measured the efficiency of XCI initiation and differentiation by means of H3K27me3 immunofluorescence (IF),

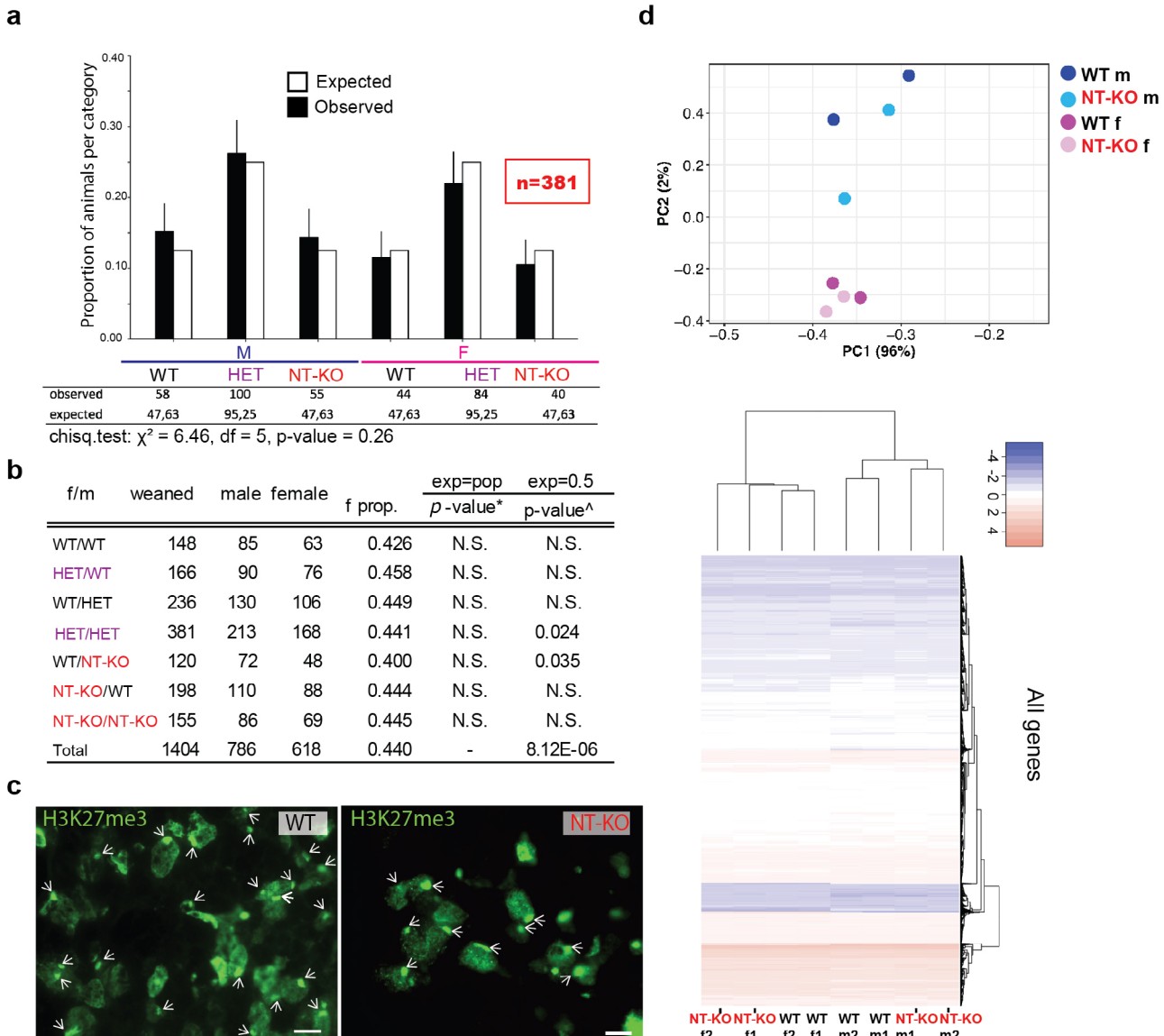

**Fig. 2 A N-term LBR deletion, does not affect XCI in vivo. a** 34 *Lbr* heterozygous (*Lbr⁺/Lbr²³⁶*, HET) mice pairs were crossed and the offspring have been genotyped by PCR analysis (see schematic in Supplementary Fig. 1a). Observed offspring genotypes indicated in black with error bars indicating 95% confidence interval (CI). A significant skewing against females is predicted in the analyzed offspring, in case Lbr plays a major role in XCI in vivo (M: male; F: female). Chi-square test analysis reveals no significant differences between expected and observed classes of male and female offspring ($X^2 = 6.46$, df = 5, *p*-value = 0.26). The number of animals used is shown in red (*n* = 381). **b** The table shows different classes of crosses and indicates the parental origin and type of the mutation. Sex ratio and binomial test analysis are also shown. Statistical tests are run vs the observed total population ratio of males/ females (m/f) and vs the expected, theoretical 50% ratio. *Two-tailed binomial test against population female proportion (0.440). ^Two-tailed binomial test against expected female proportion (0.50). **c** Xi-lamina association reveals no differences between WT and LBR NT-KO animals. H3K27me3 staining (Xi surrogate) is shown. Cross-sections from thymus tissue are shown. Arrows indicate the inactive X chromosome. Scale bar indicates 10 μm. **d** Top, RNA-Seq principal component analysis and hierarchical clustering of the LBR NT-KO vs WT are shown (number of genes: 12,791). The heatmap depicts scaled expression for better visualization. Categories are shown in the legends.

and qRT-PCR (Supplementary Fig. 4)²³. We do not observe any statistical difference between the WT and LBR NT-KO line, indicating that an equal number of cells have initiated XCI at day 5 of differentiation (Supplementary Fig. 4a, b). Similarly, we do not report major differences in the differentiation efficiency between the WT and LBR NT-KO lines (Supplementary Fig. 4c). We went on testing the localization of the Xi from the lamina at 5 days of differentiation by means of double IF. We measured the Xi-center-to-lamina (H3K27me3 domain-LaminB1) distance for the WT and NT-KO cell line. We show that the average Xi-Lamina distance is significantly larger for the LBR NT-KO line

vs the control line (Fig. 3a, b), in agreement with previous work¹¹. We went on to test whether an increased distance to the lamina also affected gene silencing, by means of qRT-PCR, on several known XCI-silenced and escapee genes²⁶,²⁷ (Fig. 3c). We find minor differences between the two lines, suggesting that while the Xi localization is mildly affected in our mutant, this does not substantially affect early gene silencing for the tested genes, as previously shown²⁴. However, we cannot exclude heterogenous silencing defects among the cell population, or more widespread deregulation using genome-wide transcriptomic analysis (i.e. RNA-seq).

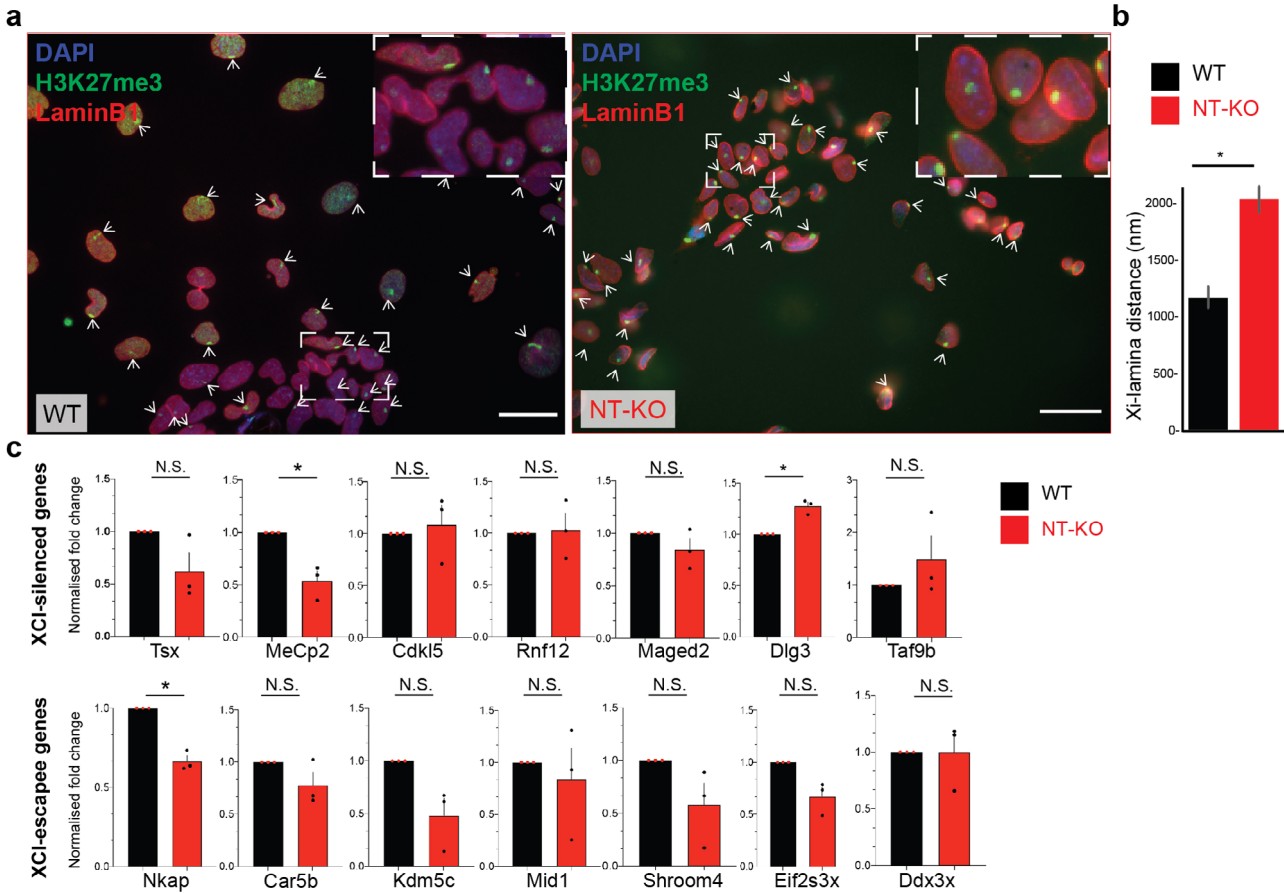

**Fig. 3 The reported LBR N-term mutation affects the Xi localization but not gene silencing in random XCI. a** Left, Xi-lamina localization is shown by means of H3K27me3/LaminB1 IF. H3K27me3, green; LaminB1, red; DAPI, blue. Dashed boxes indicate Zoom-in images in the top right corner. Arrows indicates the inactive X chromosome. Scale bar indicates ~25 μm. **b** Xi-centre-LaminB1 distance quantification ($n = 681$). Samples are color-coded as shown in the legend. Error bars represent standard deviation of the mean (SEM). **c** qRT-PCR of XCI-silenced and escapee genes[26, 27]. Analyzed genes are shown. Data from three biological samples are shown. * indicates statistical significance using a two-tailed *t*-test ($p \leq 0.05$). Error bars represent standard deviation of the mean (SEM). *p*-values: MeCp2, $p = 0.039$; Dlg3, $p = 0.020$, Nkap, $p = 0.011$. Gapdh has been used as an internal normalization control.

## Discussion

Overall, our analysis suggests that our N-terminal mutant mouse more closely reproduces the human Pelguer–Huet phenotype than currently available mouse models, which only harbor mutations in the hydrophobic domain of LBR[5,9]. This can help to further tease apart the dual functions of the *LBR* protein in development and X chromosome inactivation (XCI). In particular, this system can provide a superior understanding of the role(s) of LBR's N-terminal domains vs its sterol-reductase activities[1,10]. Our model also suggests that the N-terminal of LBR plays a role in heterochromatin organization[28] and random XCI in differentiating ESC as previously reported[11,24] but is apparently redundant for XCI functions in vivo, similar to what has been seen with *Rnf12*, *Xist*'s major activator[29]. It is, however, possible that the Xi localization and gene silencing are affected in early embryos in the absence of LBR N-terminus. XCI and developmental defects can either be fatal (~40–70% embryonic plus perinatal lethality) or compensated later in development. Several studies have reported that while *Lbr* expression is crucial in early development, its expression, is greatly reduced during development (with the exception of specialized blood cells), where lamins A/C have a more important role in the organization of peripheral heterochromatin[30]. Therefore, it is possible that stochastic mechanisms, allow for proper development[31], including XCI, in the absence of LBR N-terminus, in a fraction of the animals. These compensatory events could be happening through analogous proteins (i.e., Lamins A/C, for survival and blood development)[30] and/or alternative pathways, such as nucleolus-mediated Xi targeting to the nuclear periphery, in the context of XCI[32,33]. We should note, however, that we do not have the power to disentangle developmental and XCI-specific defects with the genetic and molecular biology tools used in this work. Single-cell-analysis in early development in both sexes can be employed to untangle the different roles of *Lbr* in development and X chromosome inactivation, in future studies.

The difference between the sexes in gene expression in LBR NT-KO vs control is also unexpected. Wichers et al., using a pericentromeric heterochromatin reporter, have shown significant differences in the efficiency of the reporter silencing between the XX vs XY animals[34]. In particular, they reported that XY animals silenced the reporter gene more efficiently than XX animals, independently of sex[34]. These differences are likely related to the sex complement and the amounts of cellular heterochromatin available (heterochromatin sink effect), between the sexes[34]. It is possible that our *Lbr* mutation affects the overall heterochromatin radial and peripheral organization[5,30,35] and, in turn, this affects gene regulation differently between the sexes[34]. Alternatively, *Lbr*, might regulate X-linked pathways, which upon deregulation could majorly affects males, phenocopying male lethal, female dominant disorders[36]. As a potential caveat, we

cannot exclude that LBR localization to the inner nuclear membrane might be affected in our mutants. However, recent reports have shown that a LBR complete N-terminal deletion is capable of nuclear-membrane localization via its transmembrane domains[21]. Considering all the data presented and the literature evidence[21,22], it is likely that the phenotype observed is largely due to the specific N-terminal deletion and the severe protein reduction[21], though further studies are needed for a complete characterization of our N-terminal deletion.

Finally, while we believe that more studies at the single-cell level are needed for understanding the role(s) of the different domains of LBR in vivo, we suggest that the presented mutation is a superior model for studying human Pelger–Huet anomaly[37] and a very valuable pre-clinical tool.

## Methods

**Generation of _Lbr_ mutant mice and crossing**. Embryo microinjection was performed as in Harns et al.[38]. Briefly, fertilized eggs from superovulated females were collected and placed in KSOM medium at 37 °C in a 5% $CO_2$ incubator. Mineral oil was placed on top to avoid evaporation. Lbr guide n2 (GTGAAGTGGTCAGAG GCCGA) and Cas9 mRNA (12.5 ng/µl) (SIGMA) were resuspended in injection buffer (1 nM Tris-HCl pH 7.5, 0.1 mM EDTA) and injected in embryos using a microinjection needle. RNA guides were prepared as previously described[39]. Injected embryos were kept in KSOM medium at 37 °C in a 5% $CO_2$ incubator for 4.5–5 days in microdrops for in vitro analysis, or transferred to foster mums after about 24 h. Mineral oil was placed on top in order to avoid evaporation. About 100 injected 2C-stage embryos were transferred into two foster mothers and left to develop to term. Only healthy 2C-stage embryos were transferred to foster mothers. In order to genotype the animals that were born (26 animals), we extracted DNA from tails using a Qiagen kit (DNeasy Blood and tissue). DNA was then subjected to PCR (LbrGenomicF1/LbrGenomicR1), Forward primer, LbrGenomicF1: 5′-CCTAAAAGCCAGGGTCCTTTC-3′, Reverse primer, LbrGenomicR1: 5′-TGTTACGTTGTCAGGGTTTAATC-3′) (Supplementary Fig. 5c). PCR bands were gel extracted and topocloned using Strata PCR cloning kit (StrataClone) and sequenced by Sanger sequencing using the vector T3/T7 primers. Sequences were aligned to the reference gene using the SnapGene program. For each mutation we checked whether it affected the proteins open reading frame, using SnapGene and online tools. PCRs were performed using a thermal cycler using the following protocol: 95 °C 3′, 94 °C 20″, 60–62 °C for 20″, 72 °C for 20–30″ (30–38X), 72 °C for 5 min. KAPA2G Fast ready mix was used (KAPA) for genotyping.

A 236 bp (c.-14-179_43del) mutation was selected as it eliminated the 1st methionine/Kozac sequence (initiation of translation) and was predicted to result in either a full KO or a truncated protein missing the Tudor and RS domains suggested to be important for interacting with chromatin/Xist RNA[11,13,23]. The male founder was screened for off-target mutations at the predicted loci (MIT sgRNA designer) and no mutations were found. The founder was subsequently backcrossed with WT C57BL/6N females. The heterozygous offspring were subsequently intercrossed, continuing the colony until sufficient offspring could be produced to robustly determine any sex or genotype biased mortality. In total, 381 offspring were reared from HET/HET crosses across three generations. Animal work received ethical approval and complied with EU (Italian Home Office, Ministero dell'Interno) and UK (UK Home Office) regulation.

**Derivation of ES cells lines from mouse models**. In order to derive the desired cell lines, we set up LBR homozygous animals ($Lbr^{236}/Lbr^{236}$) crosses and matched WT crosses. Blastocysts were isolated by flushing the uteri of superovulated female mice with M2 media at day 3.5 pc followed by incubation for 24–36 h in KSOM media. After hatching and attaching to the cell culture plate embryos were trypsinized and plated in 2i media. About a week after, ~10 clones were picked and expanded and subsequently, sexed and genotyped, and karyotyped. After extensive characterization of these clones, we decided to focus on two female NT-KO ES cell lines, and an _Lbr_ WT ES cell line.

**Differentiation of ES cells**. Cells were grown in 2i conditions as previously published[40]. For differentiation experiments, the wells of a 6-well plate or a 10 cm TC dish were pre-coated with gelatin (0.1%) containing laminin (1:1000) solution and incubated at 37 °C for 15 min. About 150,000 cells per well in 6-well plate were plated in each well in neuronal differentiation medium (NDM) as previously published[41]. The cells were differentiated for 5 days and harvested for qRT-PCR or IF analysis.

**RNA extraction qRT-PCR**. For tissue analysis, RNA has been extracted using Trizol (Invitrogen) according to the manufacturer's protocol. RNA-seq analysis was carried out as previously described[5]. For cell work, RNA was extracted using the RNaesy mini kit according to the producer's manual (Qiagen). RNA was DNAse treated for 30 min at 37 °C and DNAse was then inactivated using a cDNA

kit (Turbo DNAse, Ambion). Reverse-transcription was carried out using a kit from Thermo Scientific in accordance with the manufacturer's instructions (1–2 µg of total RNA). qRT-PCR was performed using Biorad SYBR-green reagent using an Applied Biosystem real-time system, following standard protocols. A wide panel of primer pairs was used to test cell differentiation and XCI progression, in the initial phases of the project. Selected primers were then used for all experiments. Primer used in this paper are in Supplementary Data 3.

**Immunohistochemistry and immunofluorescence**. Serial 7-µm sagittal sections were collected on slides. Hematoxylin and eosin (HE) staining was performed to confirm the quality of the tissues. Immunofluorescence staining was done for H3K27me3 (1:200; Active Motif cat. # 39536) and LaminB1 (1:500; Proteintech cat. # 66095-1-Ig) after microwave antigen retrieval using 10mM citrate buffer pH 6.0 for 20 min. No antigene retrieval was used for conventional IFs.

**Microscope acquisition and Xi-lamina distance analysis**. Leica TCS SP5 confocal microscope or Leica DM6000B microscope was used to acquire images using 40/63× objective. Images were saved in jpg or tiff format for ImageJ analysis (https://imagej.nih.gov/ij/). In brief, Xi-lamina distances were calculated using ImageJ plugins.

**RNA-seq analysis and GO analysis**. Reads were trimmed using _cutadapt_ and aligned to the genome with STAR[42]. The _FeatureCounts_ software has been used to calculate the number of reads per gene[43]. Genes annotations were taken from ENSEMBL _Mus musculus_ GRCm38.88 (https://www.ensembl.org/Mus_musculus/Info/Index). Lowly expressed genes (with <1 count per million across all groups) were filtered out. Characterization of RNA species and quality control were assessed via the Bioconductor package NOISeq[44]: 91% of the genes were protein-coding and the few species of rRNA ($n = 2$), miRNA ($n = 10$) and Mt-RNA ($n = 16$) were filtered out. Finally, differentially expressed genes analysis was performed by the Bioconductor package edgeR, using TMM normalization and a generalized linear model (glmFit). False discovery rate (FDR) was required to be below 0.05 and fold-change >2 (https://www.bioconductor.org)[45]. Gene Ontology (GO) analysis was done using gProfiler (https://biit.cs.ut.ee/gprofiler/gost) with a significance threshold of 0.05 and the biological processes (BP), molecular functions (MF), cellular components (CC) are shown. Raw and processed RNA-seq data has been deposited at GEO. Access number: GSE165447. See also Supplementary Data 1.

**MicroCT acquisitions**. Front and rear paws from WT and $Lbr^{236}/Lbr^{236}$ animals were dissected after cervical dislocation of the animals followed by exsanguination. MicroCT acquisition and analysis were done using a Micro Photonics Instrument, in accordance with manufacturer's instructions.

**Immunoprecipitation–mass spectrometry**. Spleen and small intestines were taken from wild-type and $Lbr^{236}$ homozygous mice and homogenized in RIPA buffer with a Dounce homogenizer to extract proteins. For each sample 500 µl of extracted lysate was brought to 1 ml with RIPA buffer and precleared with 50 µl Protein G Dynabeads (Thermo Scientific) for 30 min on a rotating wheel at 4 °C. Beads were removed and 15 µl of antiLBR (Santa Cruz Biotechnology, Cat sc-160482 (G14)) was added to the cleared lysate and incubated at 4 °C for an hour on a rotating wheel, after which 50 µl of washed protein G beads were added and incubated for another hour. Beads were washed three times with 250 µl RIPA buffer and eluted in 20 µl Laemmli buffer by boiling for 5 min. Samples were run on a 10% PAGE gel and analyzed by mass spectrometry.

**LC-MS/MS**. Bands were cut from the gel and subjected to in-gel digestion with trypsin (PMID: 25278616). Peptides were extracted from the gel pieces by sonication for 15 min, followed by centrifugation and supernatant collection. A solution of 50:50 water:acetonitrile, 1% formic acid (2× the volume of the gel pieces) was added for a second extraction and the samples were again sonicated for 15 min, centrifuged and the supernatant pooled with the first extract. The pooled supernatants processed using speed vacuum centrifugation. The samples were dissolved in 10 µL of reconstitution buffer (96:4 water:acetonitrile, 1% formic acid and analyzed by LC-MS/MS.

Peptides were separated using the nanoAcquity UPLC system (Waters) fitted with a trapping devise (nanoAcquity Symmetry C18, 5 µm, 180 µm × 20 mm) and an analytical column (nanoAcquity BEH C18, 1.7 µm, 75 µm × 200 mm). The outlet of the analytical column was coupled directly to an LTQ Orbitrap Velos (Thermo Fisher Scientific) using the Proxeon nanospray source. Solvent A was water, 0.1% formic acid and solvent B was acetonitrile, 0.1% formic acid. The samples were loaded with a constant flow of solvent A at 5 µL/min onto the trapping column. Trapping time was 6 min. Peptides were eluted via the analytical column a constant flow of 0.3 µL/min. During the elution step, the percentage of solvent B increased in a linear fashion. The peptides were introduced into the mass spectrometer (Orbitrap Velos, Thermo) via a Pico-Tip Emitter 360 µm OD × 20 µm ID; 10 µm tip (New Objective) and a spray voltage of 2.2 kV was applied. The capillary temperature was set at 300 °C. Full scan MS spectra with mass range 300–1700 $m/z$ were acquired in profile mode in the FT with a resolution of 30,000. The filling time was set at

maximum of 500 ms with limitation of $1.0 \times 10^6$ ions. The most intense ions (up to 15) from the full scan MS were selected for sequencing in the LTQ. Normalized collision energy of 40% was used, and the fragmentation was performed after accumulation of $3.0 \times 10^4$ ions or after filling time of 100 ms for each precursor ion (whichever occurred first). MS/MS data was acquired in centroid mode. Only multiply charged (2+, 3+, 4+) precursor ions were selected for MS/MS. The dynamic exclusion list was restricted to 500 entries with maximum retention period of 30 s and relative mass window of 10 ppm. In order to improve the mass accuracy, a lock mass correction using the ion ($m/z$ 445.12003) was applied.

**Data processing, MaxQuant**. The raw mass spectrometry data were processed with MaxQuant (v1.6.3.4)[46] and searched against the Uniprot *mus musculus* (UP000000589; 55462entries; latest update: 20200321). The data was searched with the following modifications: Carbamidomethyl (C) (fixed modification), Acetyl (N-term), and Oxidation (M) (variable modifications). The mass error tolerance for the full scan MS spectra was set to 20 ppm and for the MS/MS spectra to 0.5 Da. A maximum of two missed cleavages for trypsin/P was allowed. For protein identification a minimum of two unique peptides with a peptide length of at least seven amino acids and a false discovery rate below 0.01 were required on the peptide and protein level. Quantification was performed using iBAQ values[47] calculated as the sum of the intensities of the identified peptides and divided by the number of observable peptides of a protein.

**ORF and alternative translational start-site predictions**. Protein and RNA sequences were downloaded from UCSC genome browser and Uniprot, respectively (https://genome.ucsc.edu/; https://www.uniprot.org/). Alternative translational start-site predictions were carried out using TIS-database (TIS-db, http://tisdb.human.cornell.edu/), Netstart (http://www.cbs.dtu.dk/services/NetStart/), ATGpr (https://atgpr.dbcls.jp/) and SnapGene (https://www.snapgene.com/). Based on this analysis the M118 is predicted to be the most likely new initiation start site, although we cannot formally exclude M149, as second potential translation start site, compatible with WB and mass-spectrometry data.

**Western blot (WB) analysis**. Whole lysate cell extracts were obtained after incubation on ice for 30′ in RIPA buffer with protease inhibitors, followed by centrifugation. WB analysis was done using standard SDS-PAGE protocols from animal-derived cells. LBR antibodies (abs) were purchased from Novus biological NBP2-14185 (recognizing LBR N-term; human LBR aa 1–70), and from Santa Cruz (sc-160482, clone G14, recognizing an internal region of LBR); Vinculin ab was also bought from Sigma (V9264-200UL) (Supplementary Figs. 1 and 5a, b).

**Blood smears**. Blood was collected by cardiac puncture from heterozygote, homozygous mutant and wild-type, female and male mice, and blood smears were prepared by Giemsa staining. Images were taken with a brightfield microscope at ×63 oil immersion Leica DM6000B and white blood cells were examined for chromatin defects.

**Statistics and reproducibility**. Statistical analyses were performed using Microsoft excel, Prism v8 and R. Chi-square test and unpaired, two-sided Student's *t* test were also used where appropriate. $P < 0.05$ was considered statistically significant. Individual *p*-values are shown, where possible. All experimental data points are clearly presented in the figure legends or in the figure the statistical test used and the error bar type is clearly indicated.

**Reporting summary**. Further information on research design is available in the Nature Research Reporting Summary linked to this article.

## Data availability

Next-generation sequencing data has been deposited in GEO, access number: GSE165447. Differentially expressed genes are available in Supplementary Data 1. All source data underlying the graphs and charts presented in the main figures are available in Supplementary Data 2. Sequences for primers used in this study are available in Supplementary Data 3. Mass-spectrometry data has been deposited in the PRIDE database, accession number: PXD024111. All data is available upon request.

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

## Acknowledgements

We would like to thank Phil Avner for his incredible generosity, support and long-term mentorship. We want also to thank all the members of the Avner and the Hoffmann groups, Tom Vulliamy, Guifeng Wei, Greta Pintacuda, Giuseppe Trigiante, and Gian Tartaglia for critical reading of the manuscript. Paolo Fruscoloni and Tiziana Orisini for help with the micro-CT acquisitions. Inderjeet Dokal, Christophe Lancrin, and his group for help with the blood analysis. Andreas Buness for initial work on the data analysis. We would like to thank Mitch Guttman for initial discussion of this project and continous support. Monica Di Giacomo for initial work on this project. Olga Boruc, Danilo Pugliese, Stefano Tatti, and all the members of the EMBL-Rome animal house for continuous support toward this project. We would like to thank Frank Stein and Mandy Rettel from the EMBL Proteomic Core facility for their help with the mass-spectrometry experiments. A.C. is funded by a Rett Syndrome Research Trust (RSRT), BARTS CHARITY grants, and intramural QMUL support. This work has been co-funded by EMBL grants. This paper is in memory of Prof. Maurizio D'Esposito, an outstanding scientist, a mentor, and a friend.

## Author contributions

A.C. thought and designed this work. A.N.Y., E.P. and N.R.B. performed most of the experiments. A.C., A.H., B.M.M. and A.L. also contributed experimentally to the paper. In particular, A.C. designed the guides for the generation of the NT-KO (and other mutants), prepared the reagents for the microinjections and did the initial character-ization of the mutant lines, with technical support from A.L. A.Y. performed all the experiments/scoring/statistics shown in the paper for the selected animal model. N.B.R. performed all the experiments in differentiating ES cells (with A.C.), and all shown western blots. E.P. performed all the histology sections and helped with the tissue pre-paration for all experiments performed in the selected mouse model. N.P. performed the bioinformatic analysis and statistical analysis. A.N.Y. performed the basic statistical analysis (with N.P.). J.W.J. provided the theoretical input in the initial stages of the project. B.M.M. helped with the microscopic analysis and Lamina-Xi distance quantifi-cations. A.C. and A.N.Y. wrote the paper. T.G. and A.C. acted as senior authors for data interpretation. The final manuscript is the result of teamwork.

## Competing interests

The authors declare no competing interests.
