## [Peer Review File · Communications Biology]

Reviewers' comments:

Reviewer #1 (Remarks to the Author):

In this study, the authors report a new mouse model expressing an N-terminally truncated LBR protein, which recapitulates phenotypic changes observed in Pelger–Huët anomaly, a condition that is caused by LBR mutations. Based on a previous published study suggesting that the N-terminus of LBR may play a role in X chromosome inactivation (XCI), the authors expected to observe a sex bias. However, no bias in support of a major role in XCI was observed in litters obtained in several independent crossings. While these XCI-relevant data are negative, it is still important that they are reported to correct the literature. Overall this is a very short report and as such, the key conclusions are mostly warranted in light of the presented data. A few questions should be addressed, outlined below.

Major concerns

- 1) This knockout mouse was generated by CRISPR-mediated genome editing that deleted the Kozak sequence, leading to the utilization of Met 118 as the alternate start codon, ultimately resulting in a N-terminal truncation. Since the N-terminal moiety of LBR is critical for targeting to the inner nuclear membrane (INM) and interacting with chromatin (e.g. PMID: 7790369), it is at this point difficult to ascribe a partial loss of function to the specific deletion of a functional domain. The phenotype could merely result from a failure to deliver the truncated protein to the INM. This formal possibility should be discussed in the text or be addressed by experiment. Could the authors use the antibody that recognizes the truncated LBR in Western blots for immunofluorescence experiments to monitor LBR in NT-KO primary cells or use the antibodies for histology?
- 2) Figure 2C: It is difficult to draw any conclusions from these suboptimal images. It would be better to show confocal images with appropriate counterstains (suitable INM/lamina markers, Hoechst).

Minor concerns

- 1) Some of the Figure legends could provide more detail in describing the experiments. To stay with the example of Fig. 2C, it would e.g. be useful to state what the staining is for easier readability.
- 2) Ref. 8 (Tsai et al.) is incorrectly cited. This study did not address tissue-specific *in vivo* phenotypes. Rather, this study established an essential function of LBR for cholesterol synthesis and rationalized several disease-causing mutations, and could instead be cited in the first sentence of the concluding paragraph along with Ref. 1. "In particular this system can provide a superior understanding of the role(s) of Lbr N-terminal domains vs its sterol-reductase activities".

Reviewer #2 (Remarks to the Author):

The authors of the manuscript titled "A N-terminal truncation of the lamin B receptor (Lbr) gene recapitulates the phenotypes of the Pelger-Huet anomaly but does not affect X chromosome inactivation *in vivo*" describe the generation and characterization of a previously unpublished LBR mutation in mice. Previous mouse models seemed to have recapitulated the phenotypes of two LBR-associated syndromes, Pelger-Huet anomaly (PHA) and Greenberg dysplasia (GRBGD). The presented mouse model is supposed to present a "cleaner" model of PHA. The authors show that a subset of

phenotypes associated with GRBGD cannot be observed in their novel mouse model. Although the N-terminal RS domain of LBR has been implicated in the initiation of Xist-mediated silencing during random XCI, the authors claim to not observe phenotypes linked to defects in X-inactivation. In general, I find the observations very intriguing and interesting and this mouse model should be published. However, the manuscript is very short and, in my opinion, it would benefit from a more detailed description and presentation of the results. Given the fact that publication of this manuscript would result in a similar "controversy" for LBR as for Rnf12/RLIM with regard to its in vitro versus in vivo role for X-inactivation, thorough discussion of the results is necessary. While I can follow the authors argument that in their LBR mutant model the initiation of random XCI appears to be unaffected (male vs female lethality), concluding that there is no XCI phenotype (e.g. maintenance of silencing, reactivation of gene expression) is not supported by the data in their present form.

General comments:

1) I am afraid it would be too much to ask the authors for a stem cell model recapitulating their deletion to test the specific role of the N-terminal LBR domain in random XCI. Given the current situation and time it would take it might be unjustified to require these experiments to be included in this manuscript.

2) In general, for the discussion part it would be nice to discuss initiation and maintenance XCI phenotypes.

3) If possible, the authors should indicate how the mutation affects LBR in general (please also see Figure comments below). Does the indicated increase of truncated LBR transcription lead to gain of function of the trans-membrane/sterol-reductase domain? Where is the sterol reductase domain localized without the N-terminal domain (nuclear periphery, ER)?

4) The authors mention that they generated other LBR deletions using CRISPR/Cas9. It would be useful to show which ones were generated and why they focused on this particular deletion for their study.

5) For the quantification of phenotypes based on lethality in mouse crosses, would it be useful to list results by litter in the supplementary data?

6) The authors mention: "[...] (i.e. the mutation is maternally or paternally inherited in heterozygosity or homozygosity) [...]" Thus, crosses with homozygous mothers have been performed, excluding maternal contribution of wildtype LBR to the early embryo. This implies no imprinted or random XCI phenotype, rendering LBR dispensable for initiation of both types of XCI in vivo. Could this be discussed in more detail?

Regarding the XCI phenotype:

7) In general, I am not convinced that lack of an XCI phenotype can be demonstrated by only observing absence of female lethality and testing for sub-nuclear localization of the inactive X chromosome. Milder XCI phenotypes might have been missed by only focusing on these two aspects (see comments below). In the original publication by Chen et al., 2016 the authors suggested LBR would be necessary to induce silencing of X-linked genes such as ATRX and loss of LBR function led to "displacement/relocalization" from the nuclear membrane.

Studying sub-nuclear localization by using only immunofluorescence for the inactive X (H3K27me3) is not optimal. The active and inactive X chromosome tend to be at the nuclear periphery, correlating with the phenomenon that larger chromosomes in general are more peripheral. A qualitative analysis

is not sufficient to claim no phenotype with respect to subnuclear localization. One could have used an anti-lamin antibody to test for attachment of the inactive X to the nuclear lamina or alternatively quantified the localization using some type of DAPI segmentation (compare Figure 3 in Chen et al, 2016, where this has been done to describe the LBR phenotype using immuno-RNA FISH for Xist and Lamin B1).

8) Using immunofluorescence on thymus tissue cross-sections, the authors conclude the lack of an XCI phenotype. How representative is thymus tissue? Is LBR normally highly expressed in the thymus? RNAseq was performed on liver tissue if I understand correctly. Given the fact that LBR becomes down-regulated in most tissues during development, why have these two tissues been selected for analysis? Would there be a tissue with high LBR expression in which an XCI maintenance phenotype would be expected, rather than in tissues which do not have high levels of LBR anyway? Should the staining experiments and expression analysis be repeated on e.g. neutrophils, sorted lymphoblastoid tissues?

9) An XCI maintenance phenotype should be studied in more detail if possible. The presence of the Barr body marked by H3K27me3 is not sufficient to conclude absence of an XCI phenotype. Even if a gene is fully reactivated on the inactive X chromosome, the expression change would amount only to 2-fold change (which is the cut-off the authors chose for their RNAseq analysis). Ideally one would use non-isogenic crosses and focus on allele-specific expression changes. While I am aware that this will not be possible in this case, the authors could focus on known mouse escapee genes and normally silenced genes to see whether some of them become upregulated/reactivated exclusively in female mutants (keeping in mind that partial reactivation will not push the expression change beyond 2-fold). If such a trend is not present in males, this could be a good indication that the LBR RS domain is indeed involved in maintaining silencing on the X.

Regarding PHA:

10) The authors investigate an expression phenotype using liver tissue from homozygous NT-KO embryos. With regard to the Pelger-Huet phenotype, it could be interesting to check whether the mis-regulated genes correspond to genes known to be mis-regulated in individuals carrying the disease or in other mouse models of this disease (even the imperfect previous mouse model). This could help to understand how well this new model recapitulates the PHA phenotype.

11) Generally, it would be nice to perform a GO term analysis of mis-regulated genes to formally show immune-system gene regulation defects and enrichment for these genes.

12) How do the authors explain the huge discrepancy in the number of differentially expressed genes between males and females? Given the very low threshold for expression change 2-fold ($\log_2 > 1$) and FDR of 0.05 I am surprised to see so few genes change their expression in females (12). How do these numbers compare to previous disease models or other LBR mutant studies?

13) I am not a blood expert at all, but could the phenotype in Figure 1A be described quantitatively and not only qualitatively using FACS? I am not sure this would apply to PHA though.

14) The classical PHA phenotype appears to be defects in nuclear shape in neutrophils and redistribution of heterochromatin. If this is visible in Figure 1A please describe this in more detail in the text for non-experts.

Manuscript comments:

15) "This deletion produces, [...], a severe hypomorph, [...]"

I am not sure 'hypomorph' is the correct term to describe the truncated LBR protein here. Please stick to terms like 'truncation'.

16) "Because this mutation generates a truncated yet WT protein [...]"

A truncated protein cannot be a WT protein unless this shorter isoform already exists in the wildtype situation (e.g. by alternative splicing).

17) There is a constant shift in describing protein domains as hydrophilic and hydrophobic or transmembrane, RS, Tudor etc. Once the protein structure has been laid out in a Figure and described in the text, subsequent descriptions should be consistent.

18) Generally, for describing the phenotype I would suggest describing the more general aspects like viability, fertility, skewing of sexes (X-linked) phenotype before more specific phenotypes. But that is just a taste issue. It would however, make the first paragraph end on a more positive note, as the final message would be that the phenotype better recapitulates the human disease PHA but not Greenberg dysplasia.

Figure comments:

Figure 1:

Figure 1 and the publication would benefit from showing Fig S1 A and C albeit in different order in Figure 1. In my opinion Figure 1 should start with a gene map similar to Fig S1C. This map could have a 'zoom in' section in which Exon 2 and the deletion are shown. From the current representation a lot remains unclear. What should be highlighted: exon numbering (currently only transmembrane domains are numbered, this is confusing), main and alternative start codon, the location of the deletion, parts belonging to "hydrophilic" and "hydrophobic" domains, deletion location of the previous model for PHA and GRBGD.

A) It would be nice if the observed phenotype could be qualitatively described in words in the main text (alternatively the figure legend) for "non-blood-expert" readers of the publication. The observed blood defects described in the text should be highlighted in the figure with e.g. arrow heads.

C) Please indicate "front" and "rear" paw in the figure, maybe above the genotypes.

D) Maybe a bar chart like in Figure 2 would be easier for the reader at this point. The Table could be moved to supplementary data. Please alter all abbreviations (or if these are typos, please fix them) in the table head (ft/mt, borr, wea, peri-mort) into full words. If not, please explain every abbreviation in the figure legend. Also, please be consistent with the p-value numbering (e versus E) e.g. $3.06e-11$ versus $4.35E-03$

Figure S1:

C) I am not entirely sure how the authors come to the conclusion that the lower band in the wildtype is due to degradation while in the NT-KO it would be the result of using an alternative start codon. Could the lower band in the WT not be the result of alternative start codons/alternative splicing? The authors mention three different LBR antibodies in their materials and methods. Can the LBR antibody against the N-terminus and a second LBR antibody not be used to show that a truncated version of LBR is expressed?

It appears highly unusual to perform Western blot analysis using two primary antibodies (anti-LBR and

anti-Vinculin) at the same time. If the blot was cut or sequentially developed it would be appropriate to indicate this. If this is not the case, parallel development with two primary antibodies seems irregular.

The graphic representation of the Mass-Spectrometry analysis should be altered. Why not plot number of detected fragments (y-axis) on the full sequence (x-axis). If done in two colors for WT and KO this would be more informative and quantitative.

Figure 2:

D) Three-dimensional PCA plots could be transformed into two-dimensional plots for better visibility of the data. A cluster analysis of all genes with a color code normalized/scaled to the mean expression (or sth similar) is not very informative. Figure 2D could be part of the supplementary data. Other types of visualizations might be more informative for the reader (e.g. volcano plots indicating significantly differentially expressed genes and their names). A GO analysis could be in this position. Or e.g. a comparison with known mis-regulated genes in other PHA models.

Reviewer #3 (Remarks to the Author):

The authors have generated a mouse model that presents for Pelger-Huet anomaly without the additional defects that current mouse models display. With the newly developed model, the authors have parsed out the contributions of Lbr protein domains to various activities, including the unexpected result that the truncation of Lbr at the N-terminus has no effect on XCI. This is an important result, however the conclusions can afford to be better supported by more extensive protein characterization to confirm that the key player is the anticipated mutant of Lbr.

1. The description of the IP-MS experiment lacks details that are key in understanding how the authors arrived at the sequence map shown in Figure S1C. At first glance, this appears to be the result of a tryptic digest, however not all observed peptides terminate at lysine or arginine. As a reader, I am left to my own imagination about how the experiment was run, which requires me to make my own assumptions about the conclusions surrounding this experiment. The experimental details need to be included in the manuscript.

2. The characterization of the protein truncation would be enhanced by an intact mass measurement (e.g. MALDI-TOF, denaturing ESI) to determine the uniformity of the gene product. If this experiment is not feasible, an optimized digest (e.g. using a combination of proteases) for better sequence coverage may be better suited. The authors report a very low percentage for sequence coverage (28% for the WT and 12% in the KO) which is not sufficient as the main conclusions rest on the expected N-terminal truncation of Lbr. It is not suitable to make the conclusion that the N-terminal truncation was successful based on the absence of N-terminal peptides when digestion efficiency appears to be low.

3. Do the authors find evidence for protein products from alternative translation start sites? This question is in relation to point 2.

Reviewers' comments:

Reviewer #1 (Remarks to the Author):

In the opinion of this reviewer, the authors have addressed the concerns and the data are consistent with the original conclusions.

One minor edit: I did not find a statement in the revised main text regarding the possible problem with LBR localization (#1 of rebuttal letter). This statement should be included somewhere in the first paragraph of the main text.

I do not need to see the ms again.

Reviewer #2 (Remarks to the Author):

The authors of the manuscript titled "A N-terminal truncation of the lamin B receptor (Lbr) gene recapitulates the phenotypes of the Pelger-Huet anomaly and shows minor X chromosome inactivation defects" have taken the majority of comments from the first round of review into consideration and have addressed the majority of issues raised by the reviewers. Especially the effect of the presented N-terminal truncation of LBR on random XCI in vivo and in vitro has been addressed. Thus, I would recommend to consider the re-submitted manuscript for publication. Please find below my comments to this new version of the manuscript.

General comments:

As recommended in the first round of review the authors have extended their analysis of XCI phenotypes. The reported results contradict the observation from Chen et al., 2016 and are more in line with the results reported in Nesterova et al., 2019. The authors show in vivo and in ESCs derived from NT-KO mutants (in vitro), that the LBR NT is not required for initiation of XCI. I recommend discussing the results accordingly with a clean separation of expected XCI initiation and maintenance phenotypes.

The LBR NT supposedly binds Xist. The interaction of Xist with LBR has been proposed to be "required for Xist-mediated silencing" (Chen et al., 2016). Apparently, this interaction is not required in vivo and in vitro in the systems the authors looked at.

The only X chromosome related phenotype observed by the authors (in the systems they looked at) is a shift towards higher distances from the nuclear lamina. This change in nuclear organization is interesting, but was not linked to gene expression changes of normally silenced genes on the X in vivo and in vitro. I am not sure this change would qualify as a 'minor X chromosome inactivation defect' as XCI (i.e. gene silencing) does not appear to be affected in vitro and in vivo.

Line 111-113:

The role of LBR in the initiation of random XCI has been the subject of debate. I would recommend mentioning this here. (Nesterova, T. B. et al. Systematic allelic analysis defines the interplay of key pathways in X chromosome inactivation. Nat Commun 10, 3129 (2019))

Line 127-133/Figure 3B:

The authors claim in the text that there was no major difference in the expression of the tested silenced and escaping genes by qRT-PCR analysis. While based on Figure 3 I would agree for the majority of normally silenced genes, it appears as if there is more silencing of escaping genes such as Kdm5c (Jarid1c) or MeCP2 (both facultative escapees) in the NT-KO. Both are marked as non-

significant, though they are expressed at only 50% of WT levels with rather short error bars. This is counterintuitive as one would expect higher expression in the NT-KO situation if there was a silencing defect. There could be something interesting here with regard to escaping genes.

Line 127-133/Figure 3/Supp. Figure 4:

One thing that is commonly assessed in this type of experiment are differentiation dynamics. Comparing the fraction of cells with an inactive X or gene silencing dynamics only makes sense if it can be assured that two cell lines differentiate with similar dynamics. This can easily be done by testing expression/downregulation of pluripotency markers by e.g. qPCR.

Line 140-143:

Based on what the authors present in the results part LBR is not essential for XCI initiation in vivo. Based on the qPCR for silenced genes (Figure 3B) and the fraction of cells that initiated random XCI (Sup. Fig. 4), LBR is also not essential for initiation of random XCI in ESCs derived from NT-KO homozygous embryos. The greater distance to the lamina of the inactive X in NT-KO cells appears to be rather a secondary effect in heterochromatin organization than "the LBR playing a role in random XCI". I would recommend rephrasing that. The phenotype might be due to heterochromatin organizing more in the nuclear interior similar to what has been observed for silenced olfactory receptor genes in LBR+ vs LBR- cells (Clowney, E. J. et al. Nuclear aggregation of olfactory receptor genes governs their monogenic expression. Cell 151, 724–737 (2012)).

Line 149-157:

I strongly disagree with this discussion point. I would limit the discussion here to the role of LBR in gene regulation and development. As there is no clear XCI phenotype, I don't see the necessity to invoke XCI here, to explain homozygous "survivors". I am not sure there is a need to disentangle development and XCI, as the authors do not report a female-specific effect on viability or X-linked gene mis-regulation in females/differentiating female ESCs. Their in vivo work and the results on the ESC derived from their N-terminal LBR mutant mouse embryos clearly show that LBR NT is not required for initiation of XCI. Also, there is no clear mis-regulation of silenced genes in LBR mutant animals (RNAseq in vivo and qPCR in differentiating ESCs).

Minor comments:

Line 87/88: The authors mention the essential role of Xist for random XCI and Rnf12/Rlim in imprinted XCI in vivo. Proper Xist function on the paternal X is of course also essential for imprinted XCI.

Line 100: The authors refer to "these regulatory defects" without specifying beforehand what those are.

Line 158: N-terminus (a spelling mistake)

Figure comments:

To Figures 3A:

Maybe a 'zoom-in' to representative examples would help to illustrate the distance shift.

To Figure 3A/S4B:

Please specify the number of analyzed cells per experiment/number of experiments in the figures or figure legends, respectively.

Reviewer #3 (Remarks to the Author):

Manuscript#: COMMSBIO-20-1119B

Title: A N-terminal truncation of the lamin B receptor (Lbr) gene recapitulates the phenotypes of the Pelger-Huet anomaly and shows minor X chromosome inactivation defects

Comments: The authors have generated a novel mouse model that presents for Pelger-Huet Anomaly without additional defects associated with Greenberg Dysplasia. While I anticipate this is an important model and fills an important need, the characterization of the gene product (that is described as the key player) is still lacking.

I appreciate the author's effort to provide a description of the methods used in their mass spectrometry characterization. As I anticipated, the authors used trypsin for digestion of the truncated Lbr and subsequent proteomics analysis. I still maintain that the sequence coverage is too low (even for membrane proteins) and does not support the conclusions they have made. This result can be improved by:

1. Obtaining an intact mass measurement by MALDI or ESI. This would provide the most definitive evidence for the truncated Lbr gene product and rule out any anomalies due to heterogeneity in the anticipated gene product (e.g. proteins from alternative translation start sites)
2. Using a more promiscuous protease such as chymotrypsin in the IP-MS experiments to improve sequence coverage.

The authors have indicated that they are no longer capable of carrying out additional experiments because they ran out of suitable antibodies for immunoprecipitation, which brings me to a third recommendation:

3. Manually search the MS data for peptides that were missed by the analysis software. Based on the expected sequence of truncated Lbr, the authors can expect to generate ~33 unique tryptic peptides, however 13 of those peptides have residues below the threshold for analysis (<7 amino acids). The authors may consider going through the raw data to determine if the low sequence coverage can be attributed to "missed" data. However, my suspicion is that digestion efficiency was low considering there was opportunity to detect up to 20 unique peptides under these conditions and the plot in Figure S1 appears to indicate ~5.

The WB analysis that the authors claim to be complementary to the MS results also comes with discrepancies that don't strengthen the authors' conclusions. For example, the additional bands in the WT (using the SantaCruz antibody) have been attributed to an uncharacterized degradation product. In the NT-KO, the very faint band attributed to the truncated Lbr very closely resembles the molecular weight of the degradation product. The resolution offered by gel electrophoresis is not sufficient to classify these bands as different proteins and so without further characterization of the "degradation product" the claim that the faint band is the target protein seems to be a convenient conclusion rather than a supported one.

The authors have convincingly predicted the likely translation start sites, however the characterization of the resulting gene product is ultimately the most important result. The current WB and low sequence coverage are not adequate for the author's claims about the molecular composition of their truncation. The results have important implications for the generation of this novel mouse model however the characterization is lacking and severely undercuts the importance of the claims.